# Clustering analysis for classifying fake real estate listings

Maifuza Mohd Amin[1], Nor Samsiah Sani[1], Mohammad Faidzul Nasrudin[1], Salwani Abdullah[1], Amit Chhabra[2] and Faizal Abd Kadir[3]

[1] Center for Artificial Intelligence Technology, Faculty of Information Science & Technology, Universiti Kebangsaan Malaysia, Bangi, Selangor, Malaysia
[2] Department of Computer Engineering and Technology, Guru Nanak Dev University, Amritsar, Amritsar, India
[3] My Crib Resources, Shah Alam, Selangor, Malaysia



## ABSTRACT

With the rapid growth of online property rental and sale platforms, the prevalence of fake real estate listings has become a significant concern. These deceptive listings waste time and effort for buyers and sellers and pose potential risks. Therefore, developing effective methods to distinguish genuine from fake listings is crucial. Accurately identifying fake real estate listings is a critical challenge, and clustering analysis can significantly improve this process. While clustering has been widely used to detect fraud in various fields, its application in the real estate domain has been somewhat limited, primarily focused on auctions and property appraisals. This study aims to fill this gap by using clustering to classify properties into fake and genuine listings based on datasets curated by industry experts. This study developed a K-means model to group properties into clusters, clearly distinguishing between fake and genuine listings. To assure the quality of the training data, data pre-processing procedures were performed on the raw dataset. Several techniques were used to determine the optimal value for each parameter of the K-means model. The clusters are determined using the Silhouette coefficient, the Calinski-Harabasz index, and the Davies-Bouldin index. It was found that the value of cluster 2 is the best and the Camberra technique is the best method when compared to overlapping similarity and Jaccard for distance. The clustering results are assessed using two machine learning algorithms: Random Forest and Decision Tree. The observational results have shown that the optimized K-means significantly improves the accuracy of the Random Forest classification model, boosting it by an impressive 96%. Furthermore, this research demonstrates that clustering helps create a balanced dataset containing fake and genuine clusters. This balanced dataset holds promise for future investigations, particularly for deep learning models that require balanced data to perform optimally. This study presents a practical and effective way to identify fake real estate listings by harnessing the power of clustering analysis, ultimately contributing to a more trustworthy and secure real estate market.

Corresponding authors
Maifuza Mohd Amin,
p123261@siswa.ukm.edu.my
Nor Samsiah Sani,
norsamsiahsani@ukm.edu.my

## INTRODUCTION

In today's digital era, the proliferation of online platforms has made it easier than ever for scammers to create fake real estate listings. These fraudulent postings deceive potential buyers and renters and tarnish the reputation of legitimate property sellers and agents. Identifying these deceptive listings poses a significant challenge, as scammers have become increasingly sophisticated in their methods. For example, reputable and popular platforms such as e-Bay and Craiglist have been contaminated with fake real estate ads (*Baby & Shilpa, 2021*). This is because online media such as real estate platforms can be manipulated from various aspects. Data manipulation such as offering luxury properties at low prices and in strategic locations can attract the attention of buyers who are often looking for properties that are worth their budget. According to the Federal Bureau of Investigation's Internet Crime Complaint Center, more than 11,578 victims reported real estate fraud in 2021 alone, resulting in $350 million in damages (*Internet Crime Complaint Center, 2021*). The use of fake realtor images has also been reported (*Propsocial, 2015*). A sophisticated fraud strategy succeeds in deceiving not only buyers but also sellers. This has been a concern of many traders who want to invest in real estate (*Sanders, Johannessen & Kitch, 2021*).

One approach gaining traction in tackling this issue is clustering analysis. Clustering analysis involves grouping similar real estate listings based on various attributes such as price, location, images, description, and other relevant factors (*Xu, 2022*). Analyzing patterns within these clusters makes it possible to identify anomalies likely indicative of fake listings. Clustering helps in identifying natural clusters or segments in the data that may not be clearly visible. This can improve the results of the classification detection or prediction process especially for unsupervised data (*Lee, Wang & Huang, 2022*). For example, data mining can detect fraudulent patterns in real estate transactions, such as identity theft or fraudulent loan applications (*Wen-Hsi & Jau-Shien, 2010*).

The motivation for this study lies in the critical need to enhance the accuracy of real estate listings by employing clustering analysis techniques. However, there are several complexities associated with clustering analysis for classifying fake real estate listings. By delving into clustering methods, this study aims to develop a robust classification system capable of effectively distinguishing between genuine and fraudulent real estate listings.

Therefore, this study will utilize the K-means cluster analysis technique. This is because K-means provides a straightforward approach to clustering data, which can help identify unusual patterns that might indicate fraudulent listings. Moreover, detecting fraudulent real estate listings requires understanding the characteristics that differentiate legitimate listings from fraudulent ones. K-means provides easily interpretable clusters, allowing analysts to examine the features of listings within each cluster and identify potential anomalies or patterns indicative of fraud. Lastly, K-means is a well-established clustering algorithm with extensive research and practical applications across various domains, including fraud detection. Real estate platforms can leverage the wealth of knowledge and resources available on K-means to develop robust fraud detection systems tailored to their specific needs.

A K-means model was developed based on demographics (*i.e.*, price, built-up square feet, number of rooms, occupancy, property type, furnishing, tenure, car park, and unit type) and geographical information (*i.e.*, location, latitude, and longitude) of property listings data. The contributions of the study are listed as below:

- The study developed effective clustering algorithms to identify fake real estate listings based on real estate demographic and geographic information.
- The property listing data set used is real data obtained from local real estate companies in Malaysia. The dataset is specific to this study and has never been used before.
- Identify key attributes or elements that can be used to accurately detect fake property listings.

The source of the dataset for the study is from real estate agency. The data used is property listings data for October 2021. The amount of data includes 12,914 property listings and has 28 attributes. The data consists of text types, property demographic data, and images. The property data focuses on residences, stores, offices, commercial centers, warehouses, factories, and land. The distribution of real estate locations covers all states in Malaysia. Figure 1 shows the location of the states in Malaysia.

To ensure a seamless guide for the reader through this research, the article is structured into the following key sections: "Related Work" comprises a literature review of recent work related to fraud detection across various domains. "Materials and Methods" presents the data preparation, proposed clustering methodology, clustering model evaluation and distance measurement for the research. In "Results", a detailed analysis of experimental results, including optimum cluster value, optimum distance, visualization of clustering results, feature extraction, class labels, classification models on fake property listings and attribute ranking, were discussed. Finally, discussion and conclusions are included at the end of this article.

## RELATED WORK

Various methods have been used to eradicate fraud. From the researchers' point of view, various fraud prediction analyses have been developed to identify elements of fraud, such as email filtering, data intrusion element detection, identification of fake news, fake job ads, fake auctions, fake investments, fake social media dating, and so on (*Yan, Li & He, 2021*; *Raghavan & Gayar, 2019*; *Rezayi et al., 2021*; *Park et al., 2019*; *Samarthrao & Rohokale, 2022*; *Ali et al., 2022*; *Villanueva et al., 2022*; *Prashanth et al., 2022*; *Suarez-Tangil et al., 2020*; *Gowri et al., 2021*). All these studies use machine learning and data mining as an innovative artificial intelligence method. In data mining, clustering is used as a data exploration and analysis technique to find patterns, structures, and relationships in the data set (*Xu, 2022*). The purpose is to group data that are similar or have similar properties.

The versatility and capabilities of clustering techniques have been applied in several forms of fraud detection. Previous studies of fraud classification in various domains have used clustering methods such as K-means, Fuzzy, X means, density-based spatial clustering of applications with noise (DBSCAN) and hierarchical clustering, according to Table 1. All

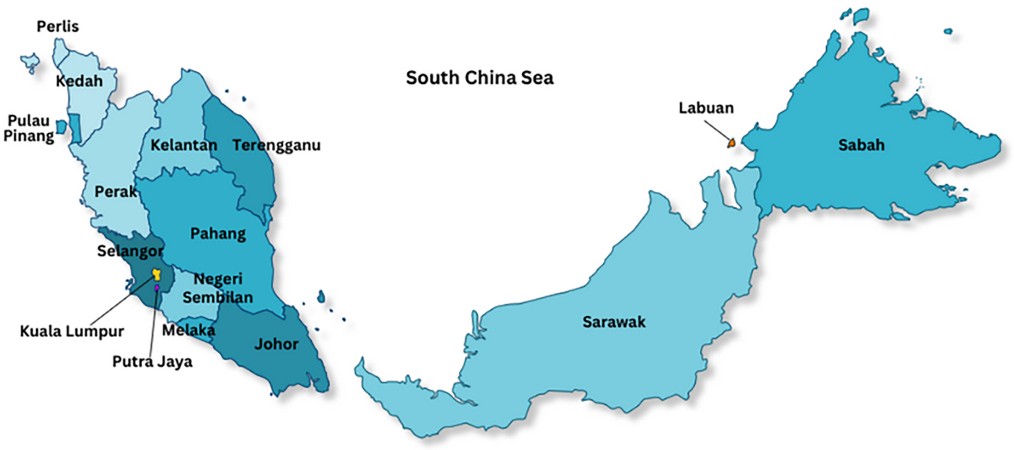

**Figure 1 Location of the states in Malaysia.**

algorithms have their own strengths and weaknesses, but they are able to produce clusters that exhibit optimal performance based on the suitability of the data used.

*Pitolli et al. (2017)* use the Balanced Iterative Reducing and Clustering using Hierarchies (BIRCH) clustering technique to present a unique solution to detect similar malware samples. The authors compare the accuracy and performance of BIRCH against other clustering algorithms. K-means, DBSCAN, and hierarchical clustering are among the clustering methods considered for comparison. Based on majority voting and machine learning methods, the authors show that BIRCH can be modified to achieve higher accuracy than or similar to other clustering algorithms. In addition, the authors present a performance comparison where BIRCH is the best.

*Phillips & Wilder (2020)* applied the DBSCAN to the content of cryptocurrency scam websites to identify different types or variants of scams. The clustering analysis identified 171 clusters of scam websites, with an average of 24 websites per cluster. The authors also used clustering to identify similarities in websites registration and ownership details. Overall, the clustering analysis was a significant contribution to this study as it helped to identify patterns and similarities in the scam websites, which could be used to better understand and combat cryptocurrency scams.

In addition, *Wen-Hsi & Jau-Shien (2010)* used the X means clustering technique to categorize fraudulent behaviour into natural groups based on fraudster's tricks. The decision tree algorithm was then employed to induce the rules of the labelled clusters. Through the analysis, fraudsters can be categorized into four natural groups based on their behaviour changes. The use of clustering algorithms to classify fraudulent behavior into clusters helps in detecting fraudsters as early as possible, making online auction fraud detection more effective.

The K-means approach was used in several studies (*Shuqin & Jing, 2019*; *Yaji & Bayyapu, 2021*; *Adewole et al., 2020*; *Eren et al., 2020*; *Motaleb et al., 2021*; *Kiruthiga, Kola Sujatha & Kannan, 2014*). All these studies achieved the goal. Research (*Yaji & Bayyapu, 2021*) for example, established a strategy to identify fake reviews by combining elements of

**Table 1 List of studies and types of clustering algorithm in fraud detection.**

| Year of study | Objective | Data | Clustering algorithm | Result |
|---|---|---|---|---|
| Wen-Hsi & Jau-Shien (2010) | To analyze fraudulent behavior changes in online auctions | Transaction data (Yahoo! Taiwan) | X means | The application of clustering techniques to categorize fraudulent behavior on natural groups is helpful in discovering fraudsters |
| Pitolli et al. (2017) | Identifying families of similar malware | Malware samples | BIRCH, K-means, DBSCAN, and Hierarchical Clustering | BIRCH obtain an higher accuracy. |
| Phillips & Wilder (2020) | To analyze public online and blockchain-based data of cryptocurrency scams. | Cryptocurrency scam reports (CryptoScamDB) | DBSCAN | The clustering analysis helped to identify patterns and similarities in the scam websites |
| Shuqin & Jing (2019) | Proposes a model for detecting fake reviews by integrating the features of comment text and user behavior | Text reviews (Yelp) | K-means | The experimental results on show that the recognition rate of the model proposed is higher than that of other single features under fusion feature conditions |
| Yaji & Bayyapu (2021) | i) Demonstrate how machine learning algorithms (K-means algorithm) can be used by an attacker to compromise data privacy. ii) Propose a defense mechanism that uses differential privacy to defend against the attack on the result. | Handwritten (MNIST) | K-means | Aggregating white noise is better than red noise in delivering defense against the result attack |
| Adewole et al. (2020) | Propose a new approach to detect spammers on Twitter based on the similarities that exist among spam accounts. | tweets texts | K-means | The study introduced a number of clusters and features to improve the performance of the three classification algorithms |
| Subudhi & Panigrahi (2016) | Detect fraudulent usage of mobile phones by analysing the user's calling behaviour using support vector machine (SVM) along with fuzzy clustering | Calls informations | Fuzzy | Fuzzy clustering and support vector machine (SVM) for detecting fraudulent usage of mobile phones is effective in identifying fraudulent calls without raising too many false alarms. |
| Eren et al. (2020) | To describe an approach to organize the literature related with COVID-19 using machine learning techniques | Scientific papers (CORD-19) | K-means | Similar documents was clustered and a web-based tool was developed for users to explore the clusters and navigate through the documents. |
| Motaleb et al. (2021) | Analyze the correlation between human abilities and individual attributes (age, gender, religion and others) to detect fake news. | Demography data of participant dan quantitative questions | K-means | The occupation and religion features had the highest F-scores and a high correlation value for K-means clustering. |
| Kiruthiga, Kola Sujatha & Kannan (2014) | To detect cloning attacks in social networks | User actions (time period and click patterns) | K-means | The clustering algorithm helps in grouping similar profiles together and identifying the cloned profiles. The paper has reported an improvement in the performance of similarity measures in finding the cloning attacks on Facebook. |

| Year of study | Objective | Data | Clustering algorithm | Result |
|---|---|---|---|---|
| *Cardoso, Alves & Restivo (2020)* | To propose novel metrics to detect two types of cheating in online courses | Log of actions performed by the students in an online course | K-means | The clustering analysis resulted in identifying pairs of accounts that have low Mean Mutual Information Rate values. |
| The proposed study | • To group similar real estate listings based on various attributes.<br>• To analyze patterns within these clusters makes it possible to identify anomalies likely indicative of fake listings.<br>• To evaluate the capabilities of the K-means model in distinguishing between fake and genuine properties using Random Forest and Decision Tree algorithms. | Durian Property real estate listings | K-means | K-means clustering successfully labelled unsupervised property listing data with excellent accuracy, reliability and validity. |

review language and user behavior. However, cluster analysis has its limitations for complex fraud tactics that do not form clear clusters. Different effective learning models, modalities, and combination techniques must be used to detect complex deception tactics. Clustering analysis is not sufficient to make a final decision about fake intent. However, clustering analyses are necessary to detect suspicious or fake data patterns. Clustering analyses can be used to form supervised data by labelling groups of data according to appropriate characteristics. Machine learning models can then be used to refine and validate the identified clusters. Therefore, a clustering analysis of unsupervised real estate data is performed in this study. The resulting data cluster labels are tested with two learning models to validate and evaluate the effectiveness of the resulting labels.

## MATERIALS AND METHODS

The proposed study's methodology for categorizing fraudulent real estate listings is depicted in Fig. 2. The two main stages of the workflow are data preparation and modelling & evaluation. Libraries from Scikit-Learn and Python were used to carry out the experiments.

### Data preparation

**Acquisition of data**. The online Durian Property listing in Malaysia provided the dataset utilized in this study. The dataset consists of 12,916 property records in October 2021 such as demographic images, geographical data, and text data of properties with 28 attributes.

**Preprocessing of data.** The main goal of this approach is to adapt the dataset before the data mining tool and the clustering algorithms can utilize it. Four pre-processing methods are shown in Fig. 2: Attribute generation, data transformation, data cleaning, and attribute selection. A statistical review of the data found some attributes with missing values such as 'NA' and −1. Attributes such as title, facing, land_D1, land_D2, postcode, and views are

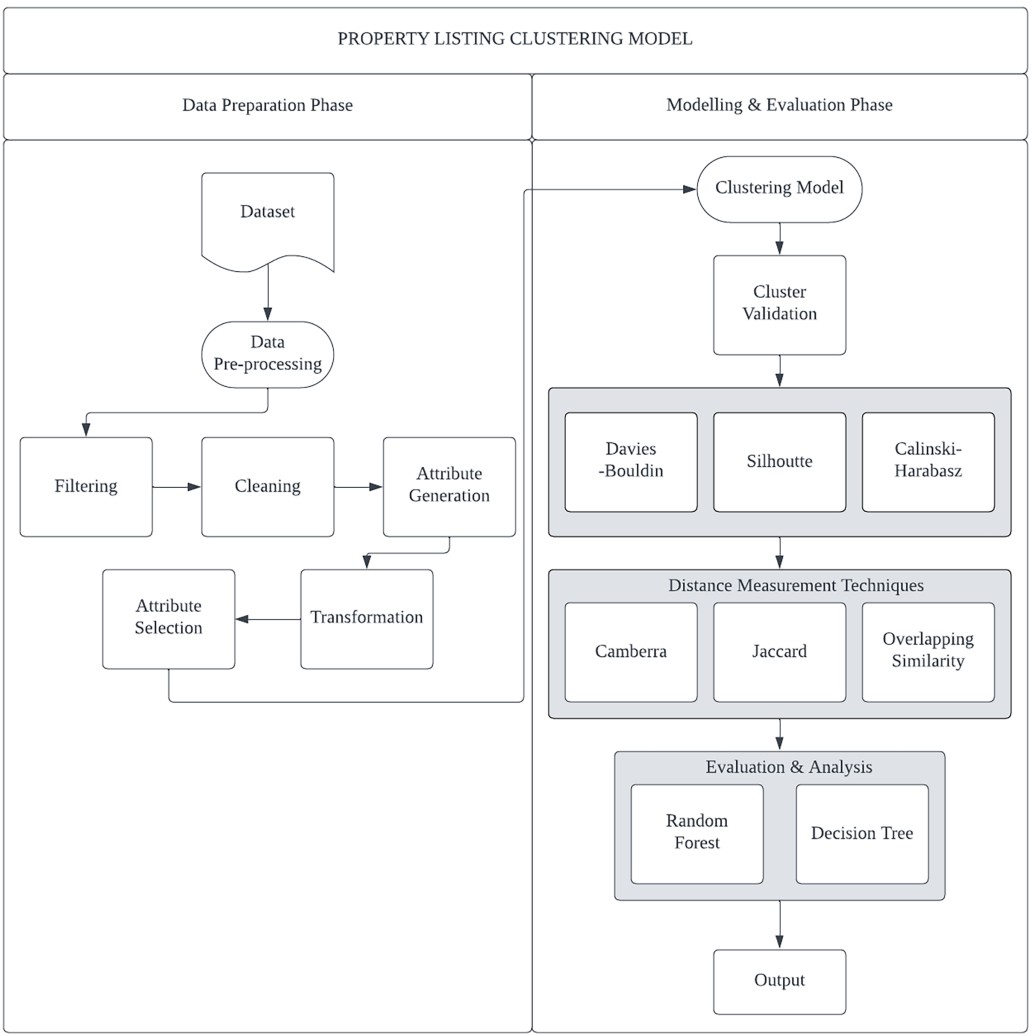

**Figure 2  The framework for the property listing clustering model.**

classified as attributes without value (contain −1 value) and were eliminated from the dataset. Whereas 84 data with missing values for almost all attributes were removed from the dataset. However, some records with missing values were replaced with values obtained from other attributes such as the Desc attribute. The Desc attribute contains text data that contains a lot of unstructured information. In order to extract the necessary information, knowledge exploration must be performed. To replace the missing values in the Car_Park attribute, the data for each housing type or Property_Type was analyzed to determine the most common Car_Park value. For example, number 2 was included as the Car_Park value for the Terrace House property because the frequency value of Car_Park is 2 for the Terrace House. The same method was applied to the Bathroom and Bedroom attributes. Three attributes were also generated: Longitude, Latitude, and Expert_Label based on the Address and Description attribute. There are only 14 attributes remaining out of the entire 28 attributes. Numerical input and output variables are often required for most machine

learning algorithms (*Adewole et al., 2020*; *Eren et al., 2020*; *Motaleb et al., 2021*; *Kiruthiga, Kola Sujatha & Kannan, 2014*). This implies that, before being input into the clustering and classification model, all attributes, including categories or nominal variables, must be converted into numeric variables. All of this is addressed through the data transformation procedure. Table 2 displays the transformation of each attribute with its description.

Models of machine learning have the ability to translate input variables into output variables. However, each variable may have a distinct domain data distribution and scale. It is possible for input variables to have multiple scales since they can have distinct units. Variations in the input variables' scale may make the modelled problem more challenging. A model that is trained with huge input values (such as a range of hundreds or thousands of units) may learn large weighting values. Large weight models are frequently unstable, which makes them perform badly during learning and make them sensitive to input values, which increases the generalization error. Therefore, the Z transform or Standard Scaler techniques were used in the property list datasets to conduct normalization. To enable comparison, the standard scaler ensures that each attribute's value falls within the same range. Every value in a data set is normalized using the Z transformation. In this case, the standard deviation is equal to one and the mean value of all values is equal to zero (*Li et al., 2021*). The following formula can be used to normalize each value in a data set using the z-transform: Eq. (1), where σ is the sample data's standard deviation, μ is the sample mean, and x j is the sample j's input value.

$$x_j = \frac{x_j - x_{min}}{x_{max} - x_{min}} \tag{1}$$

**Descriptive analysis**. After the property listing dataset is converted to numerical data, Table 3 displays the descriptive analysis of the dataset. The two attributes that have the biggest standard deviations are Price (5,098,460.11) and Built_Up_SF (64,153,663.9). Other than these two, there is an obvious gap in the other attributes.

**Feature selection**. For the purpose of feature selection, a variance threshold applied to the dataset. Variance, a metric measuring the dispersion of data within a dataset, becomes crucial in ensuring the development of an unbiased clustering model that is not skewed toward specific attributes. The selection of attributes with significant variance is essential before proceeding to build a machine learning model. High variance indicates unique attribute values or high cardinality, while low-variance attributes have relatively similar values. Attributes with zero variance exhibit identical values. Additionally, low-variance attributes, clustering close to the mean value provide minimal clustering information (*Li et al., 2021*). The variance thresholding technique, which solely evaluates the input attribute (x) without considering data from the dependent attribute, is particularly suitable for unsupervised modeling (y). Table 4 displays the variance of all attributes in property listings. Following the recommendation of *Gaurav (2019)*, a variance threshold value of 0.3 was set for attribute selection in this study to eliminate redundant features with low variance.

Upon examining the variance values for each attribute in Table 4, it is observed that all 14 attributes have variance values exceeding 0.3, indicating high behavioral patterns. All

**Table 2 List of attributes and description.**

| No. | Attribute | Values | Label | Data type |
|---|---|---|---|---|
| 2 | Price | <249,050 | 0 | Nominal |
| | | 249,050 to 300,500 | 1 | |
| | | 300,501 to 360,360 | 2 | |
| | | 360,361 to 420,500 | 3 | |
| | | 420,501 to 490,048 | 4 | |
| | | 490,049 to 584,500 | 5 | |
| | | 584,501 to 710,224 | 6 | |
| | | 710,225 to 953,000 | 7 | |
| | | 300,500 to 360,360 | 8 | |
| | | >1,804,741 | 9 | |
| 3 | Built_Up_Sf | <213.105 | 0 | Nominal |
| | | 213.105 to 320.845 | 1 | |
| | | 320.845 to 408.765 | 2 | |
| | | 4,085.7.765 to 536.380 | 3 | |
| | | >536.380 | 4 | |
| 4 | Bathroom | Numbers | - | Integer |
| 5 | Property_type | Condominium/apartment/flat/serviced residence | 0 | Nominal |
| | | Terrace house | 1 | |
| | | Link bungalow/semi-detached house/superlink | 2 | |
| | | Bungalow/detached house/villa | 3 | |
| | | Factory/warehouse/shop | 4 | |
| | | Agriculture/bungalow land/commercial/development land | 5 | |
| | | Shop office/SOHO/office | 6 | |
| | | Townhouse/cluster | 7 | |
| | | Hotel | 8 | |
| | | | 9 | |
| | | | 10 | |
| | | | 11 | |
| 6 | Occupancy | Vacant | 0 | Nominal |
| | | Tenanted | 1 | |
| | | Owner occupied | 2 | |
| 7 | Unit_type | Intermediate lot | 0 | Nominal |
| | | Corner lot | 1 | |
| | | End lot | 2 | |
| 8 | Location | Selangor | 0 | Nominal |
| | | Kuala lumpur | 1 | |
| | | Johor baharu | 2 | |
| | | Negeri sembilan | 3 | |
| | | Pulau pinang | 4 | |
| | | Perak | 5 | |

(Continued)

| No. | Attribute | Values | Label | Data type |
|---|---|---|---|---|
| | | Kedah | 6 | |
| | | Cyberjaya | 7 | |
| | | Pahang | 8 | |
| | | Melaka | 9 | |
| | | Putrajaya | 10 | |
| | | Sabah | 11 | |
| | | Kelantan | 12 | |
| | | Terengganu | 13 | |
| | | Sarawak | 14 | |
| | | Perlis | 15 | |
| 9 | Furnishing | Unfurnished | 0 | Nominal |
| | | Partly furnished | 1 | |
| | | Fully furnished | 2 | |
| 10 | Bedroom | Numbers | – | Integer |
| 11 | Tenure | Leased hold | 0 | Nominal |
| | | Freehold | 1 | |
| 12 | Car_park | Numbers | – | Integer |
| 13 | Expert_label | Not Fake | 0 | Nominal |
| | | Fake | 2 | |
| 14 | Longitude | Numbers | – | Float |
| 15 | Latitute | | | |

**Table 3 Descriptive analysis of property listing dataset.**

| | Count | Mean | Std | Min | 25% | 50% | 75% | Max |
|---|---|---|---|---|---|---|---|---|
| Price | 12,816 | 1,208,855.57 | 5,098,460.11 | 270 | 330,000 | 490,000 | 818,222 | 193,500,000 |
| Built_Up_SF | 12,816 | 712,390.207 | 64,153,663.9 | 5 | 955.75 | 1,400 | 2,395.25 | 7,229,348,280 |
| Bathroom | 12,816 | 2.7243 | 1.0695 | 0 | 2 | 3 | 3 | 5 |
| Furnishing | 12,816 | 0.6162 | 0.7126 | 0 | 0 | 0 | 1 | 2 |
| Bedroom | 12,816 | 3.5779 | 1.2453 | 0 | 3 | 3 | 4 | 14 |
| Tenure | 12,816 | 0.3850 | 0.4866 | 0 | 0 | 0 | 1 | 1 |
| Car_Park | 12,816 | 2.3068 | 0.9369 | 0 | 2 | 2 | 3 | 5 |
| Location | 12,816 | 1.3429 | 2.3364 | 0 | 0 | 0 | 2 | 15 |
| Property_type | 12,816 | 1.2175 | 1.6192 | 0 | 0 | 1 | 1 | 8 |
| Latitude | 12,816 | 3.2197 | 0.8487 | 1.2494 | 2.9619 | 3.0784 | 3.1975 | 6.5484 |
| Longitude | 12,816 | 101.7725 | 1.3088 | 92.6981 | 101.5346 | 101.6696 | 101.7643 | 118.1172 |
| Occupancy | 12,816 | 0.2652 | 0.6005 | 0 | 0 | 0 | 0 | 2 |
| Unit_type | 12,816 | 0.1955 | 0.4867 | 0 | 0 | 0 | 0 | 2 |
| Expert_label | 12,816 | 0.0452 | 0.2077 | 0 | 0 | 0 | 0 | 1 |

**Table 4 The variance values.**

| No. | Attribute | Variance |
| --- | --- | --- |
| 1 | Built_up_SF | 0.9999 |
| 2 | Price | 1.0000 |
| 3 | Bathroom | 0.9999 |
| 4 | Furnishing | 1.0000 |
| 5 | Bedroom | 1.0000 |
| 6 | Tenure | 0.9999 |
| 7 | Car_Park | 1.0000 |
| 8 | Location | 0.9999 |
| 9 | Property_type | 0.9999 |
| 10 | Latitude | 0.9999 |
| 11 | Longitude | 1.0000 |
| 12 | Occupancy: | 1.0000 |
| 13 | Unit_type | 1.0000 |
| 14 | Label | 1.0000 |

attributes' variance values fall between 0.9999 or 1.000. Consequently, all attributes such as Built_Up_SF, Price, Bathroom, Furnishing, Bedroom, Tenure, Car_Park, Negeri, Property_Type, Latitude, Longitude, Occupancy, Unit_Type, and label are selected for use in the subsequent phase.

## Proposed clustering methodology

Clustering analysis proves to be a valuable method within the realm of fraud detection as it uncovers groups or patterns within a dataset that could potentially signify fraudulent activities (*Mohamed Nafuri et al., 2022*). By grouping similar data points together, it becomes possible to identify anomalies and outliers, which might be indicative of suspicious transactions. Moreover, clustering analysis can play a pivotal role in labelling or categorizing data points for fraud detection purposes. After clusters have been established, labels can be assigned to these clusters based on recognized fraud patterns or prior instances of fraud. This procedure is commonly known as cluster labelling or cluster characterization.

The K-means algorithm is a widely chosen clustering technique due to its simplicity and easily interpretable results. It groups adjacent objects into a specified number of centroids, denoted as 'k'. The Elbow Method is a popular approach for determining the optimal number of clusters in K-means. It involves calculating the within-cluster variance (inertia) for different values of k, forming a variance curve. The optimal cluster count is often identified at the curve's initial turning point. As an alternative, the Silhouette analysis provides another way to evaluate clustering quality. This technique computes coefficients for each data point, measuring its similarity within its cluster compared to others. The silhouette coefficient ranges from [1, −1], with higher values indicating better alignment with the respective cluster.

## Clustering model evaluation

In the phase of clustering analysis, the accuracy and quality of clustering results play a crucial role in determining the algorithm's performance with the study's input data. Clustering evaluation is a distinct process, conducted after the final clustering output is generated, emphasizing its standalone nature (*Mohamed Nafuri et al., 2022*). Two methods are commonly employed to measure clustering quality: internal validation and external validation.

Internal validation involves assessing clustering in comparison to the clustering results themselves, focusing on the relationships within the formed clusters. This approach proves to be realistic and efficient, especially when dealing with real estate datasets characterized by increasing sizes and dimensions.

This particular study utilized three prevalent internal validation methods: the Davies-Bouldin index, Silhouette coefficient index, and Calinski-Harabasz index. Mathematical notations for grouping assessment measurements include D for the input data set, n for the number of data points in D, g for the midpoint of the entire D data set, P for the dimension number of D, NC for the number of groups, $C_i$ for the i-th group, $n_i$ for the number of data points in $C_i$, $C_i$ for the midpoint of the Ci group, and d(x,y) for the distance between points x and y (*Prasetyadi, Nugroho & Putra, 2022*).

The **Davies-Bouldin** (DB) metric, a longstanding and commonly employed method in internal validation measurements, utilizes intragroup variance and inter-group midpoint distance to identify the least favourable group pairs. Consequently, a decrease in the DB index value indicates the optimal number of groups. Equation (2) (*Gaurav, 2019*) defines the mathematical formula for the DB metric, highlighting its significance in assessing clustering quality.

$$\text{DB} = \frac{1}{NC} \sum_i \max_{j \neq i} \frac{\frac{1}{n_i} \sum_{x \in c_i} d(x, c_i) + \sum_{x \in c_j} d(x, c_j)}{d(c_i, c_j)} \tag{2}$$

**Index of Silhouette Coefficients.** The Silhouette Coefficient Index assesses the quality and cohesion of a group. A high Silhouette Coefficient value signifies a superior model, indicating that an object is well-matched within its cluster and distinctly differs from adjacent clusters. Equation (3) outlines the calculation for the Silhouette Coefficient of an individual sample:

$$s = \frac{1}{NC} \sum_i \left( \frac{1}{n_i} \sum_{x \in C_i} \frac{b(x) - a(x)}{\max[b(x), a(x)]} \right) \tag{3}$$

where $a(x) = \frac{1}{n_i} \sum_{y \in C_i, y \neq x} d(x, y)$, and $b(x) = min_{j \neq i} \left[ \frac{1}{n_j} \sum_{y \in C_j} d(x, y) \right]$

S ignores $c_i$ and g and calculates the density a(x) using pairwise distances between all objects in the clusters. At the same time, b(x) calculates separation, which is the average

distance between objects and alternative groupings or the nearest second group. The silhouette coefficient values in Eq. (2) can range between −1 and 1. The stronger the positive value of the coefficient, the more likely it is to be classified in the correct cluster. Elements with negative coefficient values, on the other hand, are more likely to be placed in the incorrect cluster (*Mohamed Nafuri et al., 2022*; *Kumar Hemwati Nandan et al., 2020*).

The **Calinski-Harabasz** (CH) index concurrently assesses two criteria by considering the average power-added result between two groups and the average yield of two plus forces in the group. In the formula, the numerator signifies the degree of separation, representing the extent to which the midpoint of the group is dispersed. Conversely, the denominator reflects density, illustrating how closely objects in the group gather around the midpoint. Equation (4) defines the mathematical formula for CH:

$$CH = \frac{\sum_i d^2(c_i, g)/(NC - 1)}{\sum_i \sum_{x \in C_i} d^2(x, c_i)/(n - NC)} \tag{4}$$

## Distance measurement

K-Min uses a distance-based measure to calculate the similarity value of each data point to the centroid. The minimum distance between the data points and the centroid is the most optimal value. Therefore, distance calculation plays a very important role in the clustering process (*Prasetyadi, Nugroho & Putra, 2022*). This technique is needed to determine how the data are related to each other and how they are different or similar to each other (*Kumar Hemwati Nandan et al., 2020*). This study uses three techniques to calculate the distance: Overlapping similarity technique, Camberra distance and Jaccard distance.

**Overlapping Similarity,** often known as the Jaccard similarity coefficient, calculates the degree of similarity between two sets based on the size of their intersection to the size of their union. It is primarily used to compare collections that include category or binary attributes (*Arshad, Riaz & Jiao, 2019*). The mathematical formula is defined as Eq. (5):

$$O(A, B) = |A \cap B|/\min(|A|, |B|) \tag{5}$$

**Camberra distance** is a measure of dissimilarity between two vectors or sets of numerical values. It calculates the normalized sum of the absolute differences between the corresponding elements of the vectors. Camberra distance is sensitive to small differences and is suitable for data with a wide range of magnitudes. It is commonly used in multidimensional scaling and clustering algorithms (*Rahman et al., 2021*). The mathematical formula is defined as Eq. (6):

$$D(A, B) = \sum (a_i - b_i)/(a_i + b_i) \tag{6}$$

**Jaccard distance** is a measure of dissimilarity between two sets based on the ratio of their difference to their union size. It complements the Jaccard similarity coefficient by quantifying dissimilarity rather than similarity. To perform an accurate cluster analysis,

the properties of each approach must be matched to the research data set (*Bahmani et al., 2015*). Equation (7) defines the mathematical formula:

$$J(A, B) = |A \cap B| / |A \cup B| \tag{7}$$

## RESULTS

### Optimum cluster value

This section shows the clustering results produced by using the proposed approach. The following is a guide to choosing the best group value based on a specific group evaluation matrix such as: A low Davies Bouldin index value, a Silhoutte coefficient index with a high positive value, and a Calinski Harabasz index with a high value indicate the achievement of optimal grouping. Table 5 summarizes the index value for clusters 2 to 9 based on several cluster evaluation techniques. Based on the index value in the table, it was found that cluster 2 is best for the Silhoutte and Calinski-Harabasz methods. While cluster 7 is for Davies Bouldin technique. However, cluster 2 is preferred because it was found to be the majority best in two of the three techniques performed.

Silhouette plots are further analyzed to determine the optimal group number. As seen in Fig. 3, the plots in the image clearly show that all clusters are above the average value line of the Silhouette coefficient. The Silhouette coefficient index value for group 2 is 0.2621 (referring to the red dotted line in the figure). Furthermore, there is no fluctuation in the Silhouette plot for cluster 2, and all clusters have positive score values, indicating that almost all data have been divided into the right clusters. As opposed to other cluster values (clusters 3, 4, and 5), there is a fluctuation in the Silhouette plot. Based on the analysis of the findings, it was found that the value of cluster 2 is the best for this study.

### Optimum distance

Distance calculation plays an important role in the clustering process. This technique is needed to identify how the data are interrelated, and how the data are different or similar to each other (*Alijamaat, Khalilian & Mustapha, 2010*). This study uses three distance calculation techniques, namely the overlapping similarity technique, Camberra distance and Jaccard distance. Table 6 is the distribution of data in groups for each technique. It was found that the Camberra Distance measurement technique criteria can provide accurate analysis based on non-linear real estate data conditions and is concerned with the selection of appropriate features.

### Visualization of the clustering results

The principal component analysis (PCA) method was used to reduce the dimension into a two dimensional space for further investigation (*Abdulkareem et al., 2021*). In this research, a PCA diagram was produced to visually provide clustering findings. Figure 4 illustrates three PCA plots based on different distance measurement techniques: (a) Camberra distance, (b) Jaccard distance, and (c) overlapping similarity. Data plots are colored in different colors based on clusters.

**Table 5 The index value for clusters 2 to 9 base on several cluster evaluation techniques.**

| Cluster | Silhoutte | Calinski Harabasz | Davies Bouldin |
|---------|-----------|-------------------|----------------|
| **2** | **0.262115** | **4,223.008759** | 1.611883 |
| 3 | 0.220882 | 3,596.561107 | 1.607183 |
| 4 | 0.223442 | 3,269.488797 | 1.629521 |
| 5 | 0.210306 | 2,914.352834 | 1.671145 |
| 6 | 0.216796 | 2,697.584617 | 1.518646 |
| 7 | 0.224543 | 2,470.948611 | **1.482036** |
| 8 | 0.208890 | 2,336.522510 | 1.553491 |
| 9 | 0.201820 | 2,236.680596 | 1.582852 |

Note:
Values in bold represent the best scores for each technique (Silhouette, Calinski-Harabasz, and Davies-Bouldin). Cluster 2 is identified as the best cluster based on the evaluation techniques Silhouette and Calinski-Harabasz.

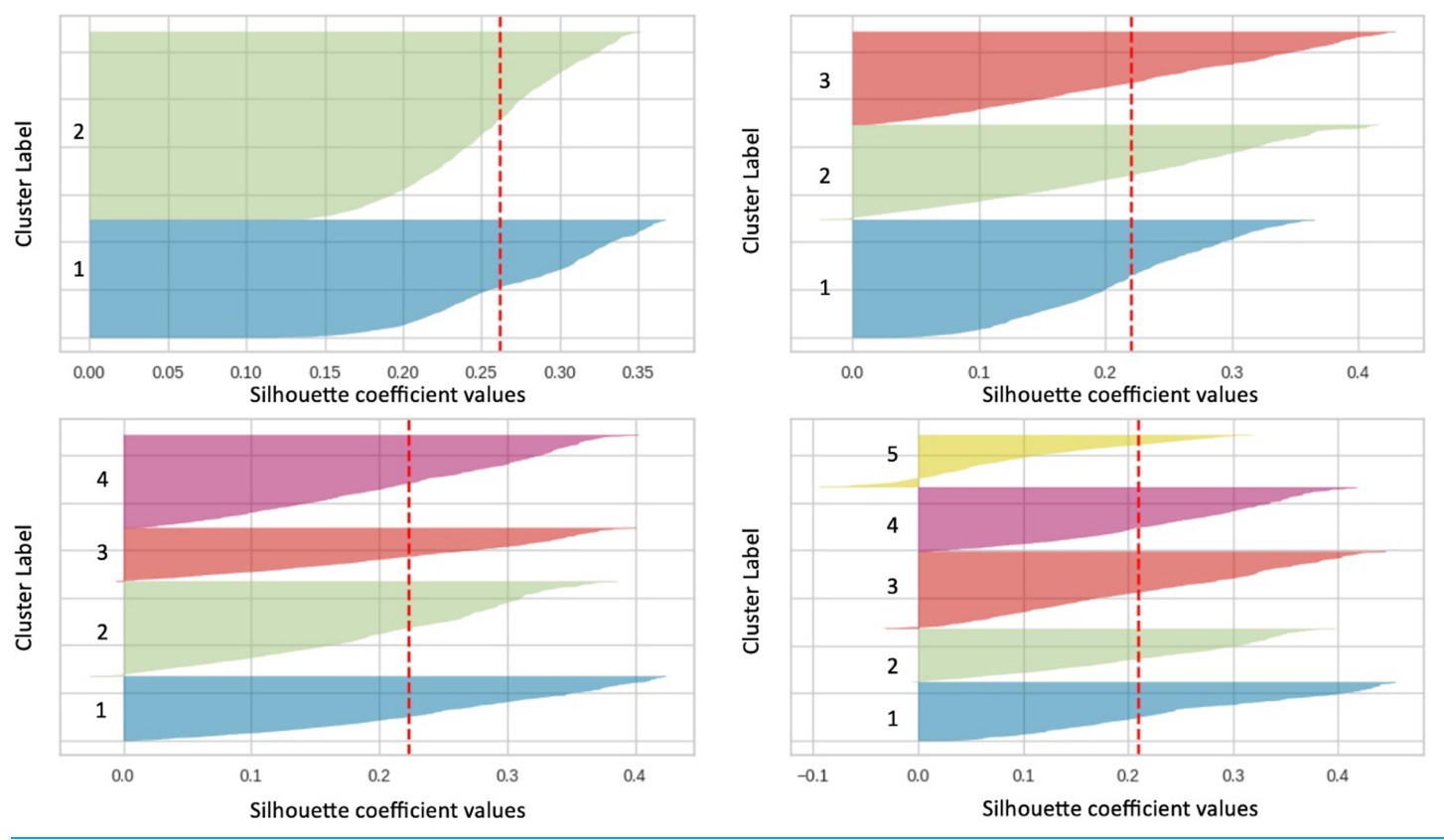

**Figure 3 Silhouette plot for clusters 2, 3, 4 and 5.**

The inter-cluster separation technique based on Camberra distance is the best because the clusters are well separated from each other based on the differentiated color, followed by Jaccard distance and Overlap Similarity. Camberra distance's preference is based on its ability to handle datasets with varying attribute scales, outliers, and sparsity while promoting effective cluster separation. Its behavior aligns with the characteristics of such

**Table 6 Distribution of data according to clusters based on several distance measurement techniques.**

| Distance measurement | K = 2 |
|---|---|
| Overlapping similarity | K0 = 3,299 |
| | K1 = 9,517 |
| Camberra distance | K0 = 7,276 |
| | K1 = 5,540 |
| Jaccard distance | K0 = 9,267 |
| | K1 = 3,549 |

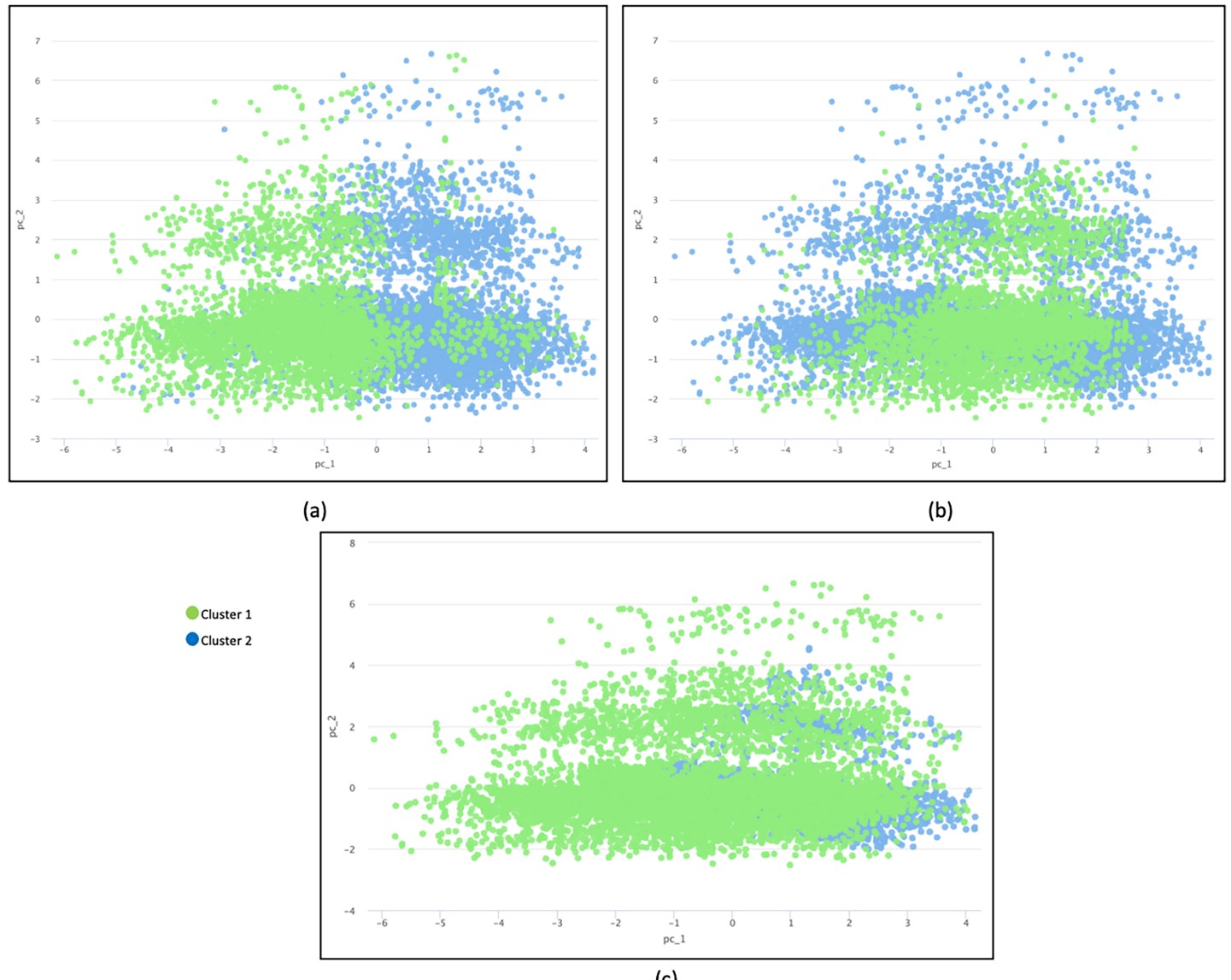

(a)

(b)

(c)

**Figure 4 Three plots of different distance measurement techniques.** (A) Camberra distance, (B) Jaccard distance and (C) overlapping similarity. Data are plotted in different colors according to clusters.

**Table 7 Descriptive statistic from the clustering results for the property listing.**

| Attribute | | Average & mode value | Cluster | |
|---|---|---|---|---|
| | | | 1 | 2 |
| | | 12,816 | 7,276 | 5,540 |
| Price | | 1,208,855.57 | 1,327,898.39 | 1,052,509.80 |
| | | | >harga purata | <harga purata |
| Built_up_SF | | 712,390.2069 | 138,896.663 | 1,465,592.197 |
| | | | <luas purata | >luas purata |
| Bathroom | | 3 | 3 | 3 |
| Bedroom | | 4 | 4 | 4 |
| Car Park | | 2 | 2 | 2 |
| Latitude | | 3.219665465 | 3.22298919 | 3.215300223 |
| Longitude | | 101.7725138 | 101.765882 | 101.7812242 |
| Frequency of category data | | | | |
| Furnishing | Unfurnished | 6,657 | 3,533 | 3,124 |
| | | 52% | 28% | 24% |
| | Partly furnished | 4,421 | 2,612 | 1,809 |
| | | 34% | 20% | 14% |
| | Fully furnished | 1,738 | 1,131 | 607 |
| | | 14% | 9% | 5% |
| Tenure | Freehold | 7,882 | 4,268 | 3,614 |
| | | 61.5% | 33.3% | 28.2% |
| | Leased hold | 4,934 | 3,008 | 1,926 |
| | | 38.5% | 23.5% | 15.0% |
| Location | Selangor | 7,852 | 4,418 | 3,434 |
| | | 61.3% | 34.5% | 26.8% |
| | Kuala lumpur | 1,476 | 836 | 640 |
| | | 11.5% | 6.5% | 5.0% |
| | Johor baharu | 759 | 442 | 317 |
| | | 5.9% | 3.4% | 2.5% |
| | Others | 2,729 | 1,580 | 1,149 |
| | | 21.3% | 12.3% | 9.0% |
| Property type | Condominium/apartment/serviced residence | 5,309 | 2,964 | 2,309 |
| | | 41.4% | 23.1% | 18.0% |
| | Terrace house | 4,353 | 2,480 | 1,873 |
| | | 34.0% | 19.4% | 14.6% |
| | Link bungalow/semi-detached house/superlink | 1,195 | 695 | 500 |
| | | 9.3% | 5.4% | 3.9% |
| | Others | 1,959 | 1,137 | 822 |
| | | 15.3% | 8.9% | 6.4% |

(Continued)

| Attribute | | Average & mode value | Cluster | |
|---|---|---|---|---|
| | | | 1 | 2 |
| | | 12,816 | 7,276 | 5,540 |
| Occupancy | Vacant | 10,479 | 5,940 | 4,539 |
| | | 81.8% | 46.3% | 35.4% |
| | Tenanted | 1,275 | 729 | 546 |
| | | 9.9% | 5.7% | 4.3% |
| | Owner occupied | 1,062 | 607 | 455 |
| | | 8.3% | 4.7% | 3.6% |
| Unit type | Intermediate lot | 10,820 | 6,133 | 4,687 |
| | | 84.4% | 47.9% | 36.6% |
| | Corner lot | 1,486 | 842 | 644 |
| | | 11.6% | 6.6% | 5.0% |
| | End lot | 510 | 301 | 209 |
| | | 4.0% | 2.3% | 1.6% |
| Expert label (text analysis) | Fake | 579 | 308 | 271 |
| | | 4.5% | 2.4% | 2.1% |
| | Not fake | 12,237 | 6,968 | 5,269 |
| | | 95.5% | 54.4% | 41.1% |

datasets, making it a valuable choice for clustering analysis. The Overlapping Equation is the weakest because it cannot separate the data points well, resulting in significant overlap. The presence of noise makes the cluster representation less accurate.

## Features extraction

Based on the results of cluster analysis in Table 7, two data clusters were formed, namely Cluster 1 and Cluster 2. The significant difference between these two clusters is in terms of price and built-up area. It was found that the real estate price and built-up area of Cluster 1 are above average with a value of MYR 1,327,898.39 and a built-up area of 1,052,509.797 square feet. The property price of Cluster 2 is below the average value of MYR 1,052,509.80 and has a built-up area of 1,465,592.197 square feet. The main distribution of property locations for these two clusters is around Selangor and Kuala Lumpur, namely 41% for cluster 1 and 31.8% for cluster 2. The frequency of distribution for both clusters is the same for the number of bedrooms which is 4, 3 for bathrooms and two for parking spaces. Most property types in both clusters are condominiums, apartments, and service apartments, accounting for 23.1% in Cluster 1 and 18% in Cluster 2. The type of real estate unit is intermediate land, representing 47.9% in Cluster 1 and 36.6% in Cluster 2. Real estate ownership is also freehold, comprising 33.3% in Cluster 1 and 28.2% in Cluster 2. Properties in both clusters are not fully furnished, with rates of 28% for Cluster 1 and 24% for Cluster 2. Additionally, the properties in both clusters are either ready for occupancy or not yet occupied, with percentages of 46.3% in Cluster 1 and 35.4% in Cluster 2. Figures 5A–5C show a bar chart of property listings by clusters for each attribute.

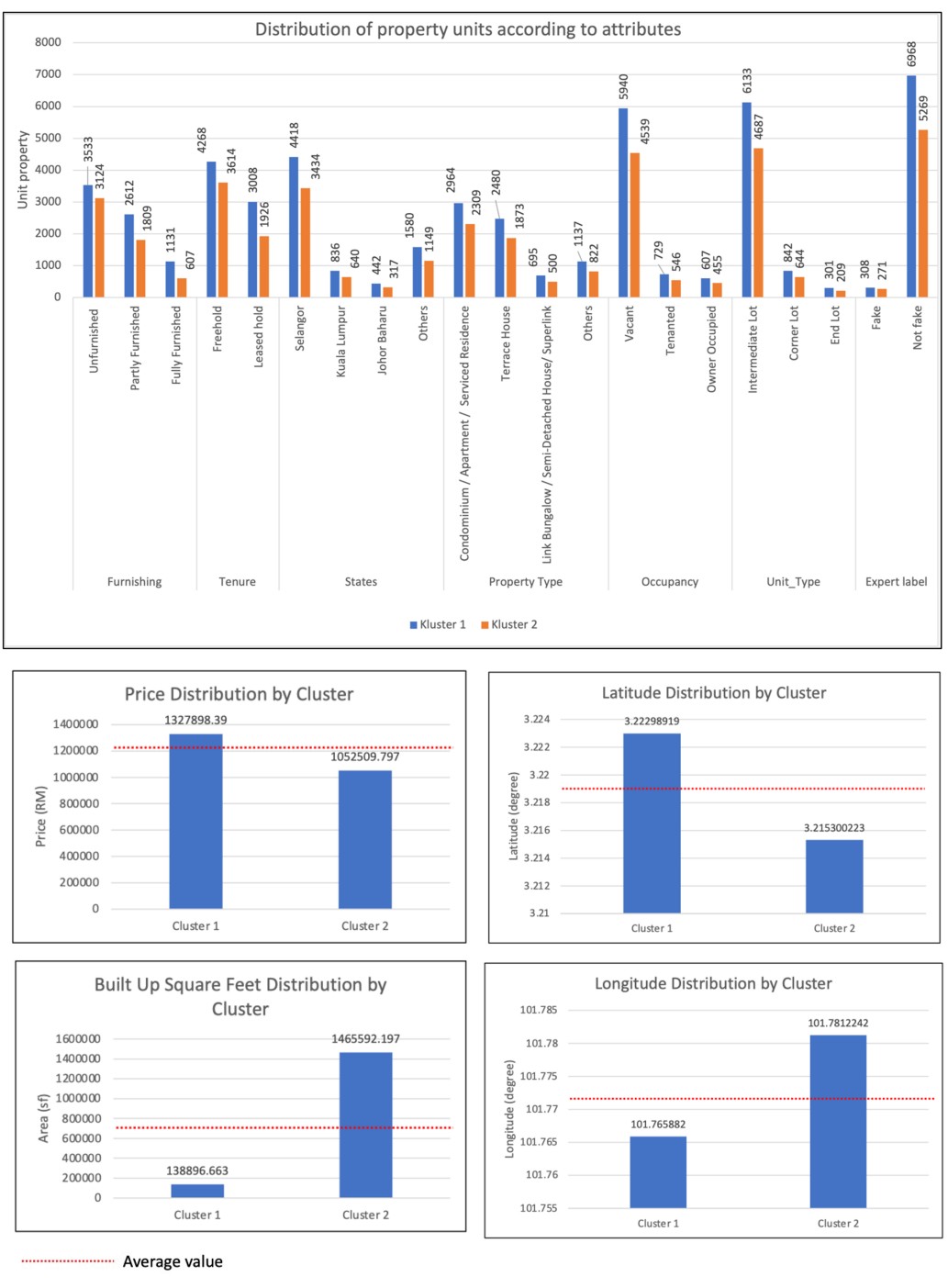

**Figure 5 Bar charts of property listings by clusters for each attribute.**

## Class label

Based on the analysis carried out, it can be concluded that clustering has created a balanced two class labels: cluster 1 (not false) with 7,276 data and cluster 2 (false) with 5,540 data. This represents a significant enhancement, given that a balanced class of data not only

**Table 8 Clusters class labels based on the property listing.**

| Cluster | Class label | No. of data | % |
|---|---|---|---|
| 1 | Not fake | 7,276 | 56.8% |
| 2 | Fake | 5,540 | 43.2% |

contributes to improved model performance but also addresses issues related to bias, ensuring fair and accurate predictions across different classes. Table 8 displays the number and percentage of property listing data that has been divided by cluster. Most of the properties in both clusters are located in the states of Selangor and Kuala Lumpur. Therefore, the comparison of the property clusters can be done based on the characteristics of other properties. Properties from Cluster 1 have a higher price than the average market value and a smaller built-up area than the average. Even with prices higher than this average, the main type of properties offered are condominiums/apartments/serviced residences. The main type of property is the intermediate property with ownership status, not yet furnished and not yet occupied.

Unlike the properties in Cluster 2, the value of the offered property is low because the owner is offered an additional bonus, namely a built-up area larger than the average value for properties in this location. The main properties in Cluster 2 are also condominiums, apartments and serviced apartments with ownership status in intermediate forms. The properties are unfurnished and unoccupied. Overall, the properties from Cluster 2 are more profitable as they offer lower prices and more spacious constructions. The real estate units are also freehold and still unoccupied. However, these particular characteristics are likely a tactic to trap buyers. In addition, the percentage of non-faults for properties in this group is much lower than that for properties in Group 2. It is impossible for the seller to sell a valued well unit below the market price because the property is located in a state with a high population density and a high demand for real estate. Therefore, properties in this group of Cluster 2 are considered likely to be Fake.

## Classification model on fake property's listing

After labelling the clusters, we proceeded to conduct further analysis by comparing the data labelled using the K-means method with the data labelled by experts. This comparison was carried out through the utilization of two distinct classification models (Random Forest and Decision Tree). This analysis aimed to ascertain the K-means model's ability to effectively distinguish between fake and genuine properties in contrast to the labels provided by experts. The results presented in Table 9 offer a detailed analysis of the performance of two classification models, namely Random Forest and Decision Tree, in the context of data labelled by experts and clusters as either fake or not fake. Notably, when we consider the accuracy metric, we observe that the Random Forest model exhibits a higher accuracy rate of 0.9618 for data labelled by clusters, surpassing its accuracy of 0.9446 for expert-labelled data. Similarly, the Decision Tree model also displays superior accuracy for data labelled by clusters (0.9571) as compared to expert-labelled data (0.9064).

**Table 9 Performance results of the classification models.**

| Model | Evaluation matriks | Expert label | Cluster label | | Paired t-test (p = 0.01) |
|---|---|---|---|---|---|
| | | | Train set (0.8) | Test set (0.2) | |
| Random forest | *Accuracy* | 0.9446 | 0.9995 | **0.9618** | Model 1 *vs.* Model 2: |
| | *Precision* | 0.0435 | 1.000 | **0.9611** | t-statistik: 120.890 |
| | *Recall* | 0.0083 | 0.9989 | **0.9488** | *p* value: 0.0001 |
| | *F1 score* | 0.0139 | 0.9994 | **0.9549** | The difference is statistically significant |
| Decision tree | *Accuracy* | 0.9064 | 1.000 | **0.9571** | Model 1 *vs.* Model 2: |
| | *Precision* | 0.024 | 1.000 | **0.9440** | t-statistik: 283.756 |
| | *Recall* | 0.0248 | 1.000 | **0.9561** | *p* value: 0.0001 |
| | *F1 score* | 0.0244 | 1.000 | **0.9500** | The difference is statistically significant |

**Note:**
The bold values refer to the performance evaluation on the test dataset. The evaluation values on the dataset are used to show the performance of the clustering method.

However, the key point of interest lies in the statistical significance of these differences, which was assessed through paired t-tests. The remarkably low p-values ($p = 0.0001$) in both cases indicate that the disparities in accuracy between the models are indeed statistically significant, suggesting that these variations are not due to random chance.

Based on the table, it was found that the precision (0.0435), recall (0.0083), and F1 score (0.0139) of the expert labeling data were very low for both models. This is due to the unbalanced expert labeling data. It turns out that the expert designation data set has a very unbalanced distribution of data between the fake (579) and non-fake (12,237) classes. This leads to a bias of the model towards the majority class and thus poor performance in the minority class (*Qi & Luo, 2022*). Traditional metrics such as precision and recall are inherently sensitive to class imbalance. If the model misclassifies minority class data, precision and recall will be affected (*Arshad, Riaz & Jiao, 2019*). However, clustering overcame the problem and was found to significantly increase the value of precision (0.9611) and recall (0.9488) as compared to before. Overfitting is not a concern for this dataset, as the accuracy, precision, recall, and F1-score values show no significant difference between the training and test datasets.

## Attribute ranking

The impact of attributes on the cluster and expert labels classification model was analyzed using summary plots SHapley Additive explanations (SHAP). This helps to understand the impact of different attributes on the model predictions.

From the SHAP plot for the cluster labels in Fig. 6, the most important attributes that influence the classification model are 'Bathrooms', 'Price', 'Built_Up_SF' and 'Car_Park'. Expert label, tenancy, occupancy, and condition are attributes that have less influence. The tendency of the attributes to make false and non-false predictions can also be analyzed by this plot. The observation results show that the attributes 'Price', 'Built_Up_SF', 'Car_Park', 'Unit_Type' and 'Occupancy' tend to make false predictions. The attributes that tend to have no false predictions are 'Tenure' and 'Expert_Label'. Otherwise, 'Bathrooms', 'Latitude', and 'Longitude' tend to be neutral.

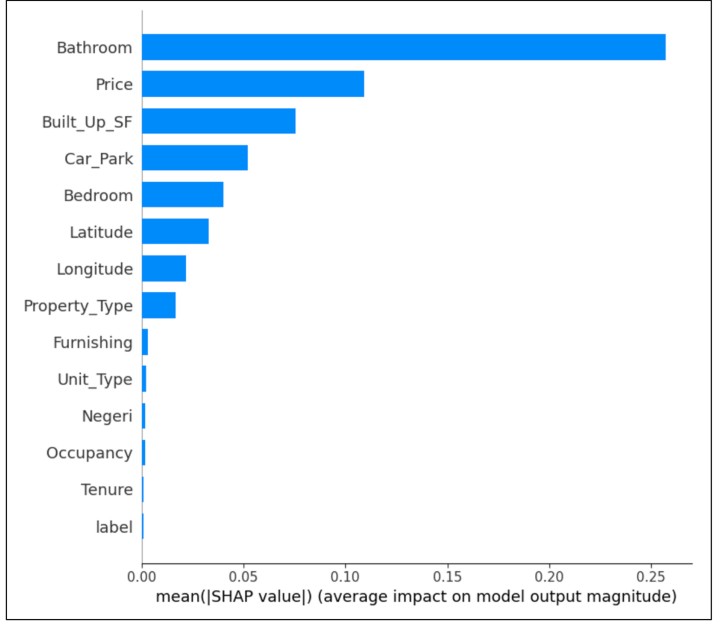
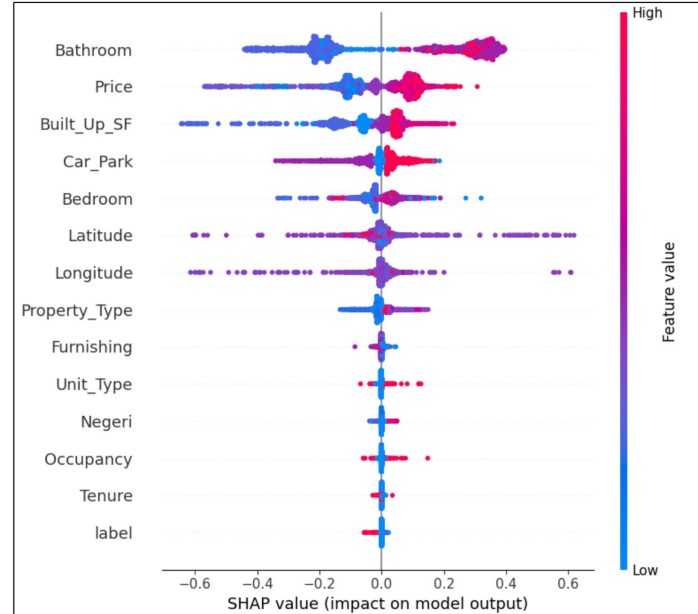

**Figure 6 Ranking of attributes in cluster label classification.**

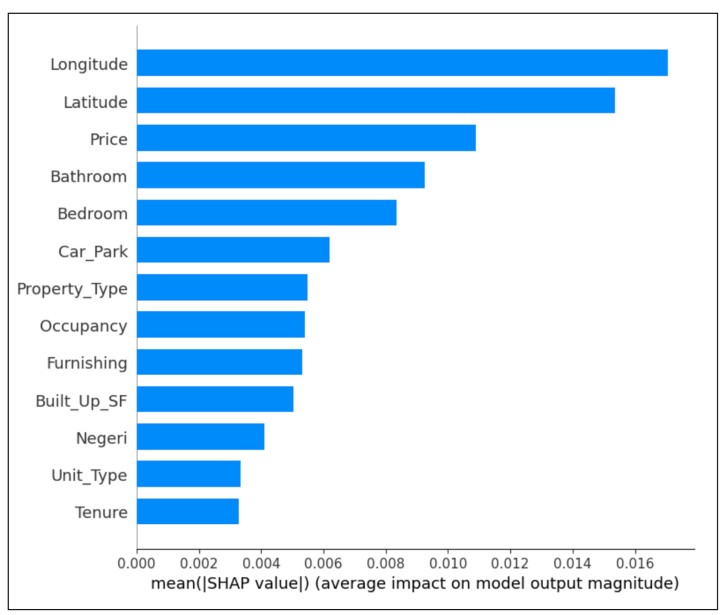
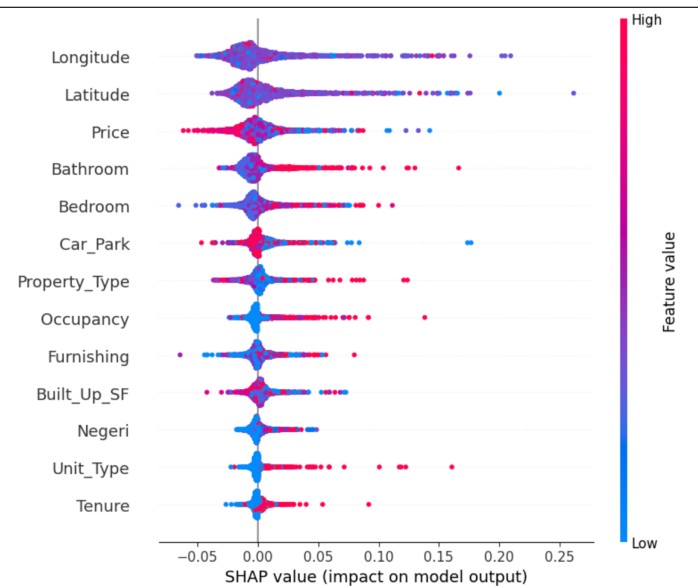

**Figure 7 Ranking of attributes in expert label classification.**

In the SHAP diagram for the expert dataset in Fig. 7, the most important attributes that influence the classification model are 'Longitude', 'Latitude', 'Price', 'Bathroom' and 'Bedroom'. Attributes that have less influence are 'Tenure', 'Unit_Type', 'State' and 'Built_Up_SF'. This plot can also analyze the tendency of attributes to make false and non-

false predictions. The observation results show that the attributes 'Bathroom', 'Bedroom', 'Property_Type', 'Unit_Type', 'Tenure' and 'Occupancy' tend to make false predictions. The attributes that tend not to make false predictions are 'Price', 'Built_Up_SF' and 'Car_Park'. In addition, latitude and longitude tend to be neutral.

## DISCUSSION

A K-means clustering analysis was conducted to classify unsupervised fake property listings using the Malaysia property dataset, Durian Property Listing. Through the analysis of this study, it was found that K-means Clustering successfully labelled unsupervised property listing data with excellent accuracy, reliability and validity. K-means was likely chosen for its scalability, ability to work with unsupervised data, clustering based on distance, flexibility, speed, and interpretability (*Rahman et al., 2021*). These characteristics make it a robust choice for classifying property listings, including detecting fake ones, and can lead to excellent accuracy, reliability, and validity of the results (*Bahmani et al., 2015*).

There are two different groups formed in the analysis of this study. The first group is classified as non-fake, while the second group is classified as fake. The first group consists of real estate listings with characteristics that meet some of the criteria associated with valid real estate listings, such as realistic real estate listings at reasonable prices. The second group contains property listings with characteristics that contradict standard property criteria, such as below-average prices for prime locations and confusing property details. The performance of the group is evaluated using two learning models, namely Random Forest and Decision Tree. It was found that the accuracy, reliability and validity of the obtained group classification are very good as compared to the expert label obtained by extracting certain words from the property description features.

However, the evaluation is still not sufficient to be used as a final decision for a more accurate classification of fake real estate listings. This is because this analysis focuses on only one type of modality, namely demographic real estate data from a single data source (*Holliday, Sani & Willett, 2018*). Several other modalities can be included in the analysis, such as images and text describing properties. In addition, various combination techniques and effective learning models should also be used in the analysis to achieve a more holistic and effective result (*Abdulkareem et al., 2021*; *Shamsuddin, Othman & Sani, 2022*). Therefore, it is proposed to use the dataset of real estate listings labelled in this study to develop a more effective model for detecting fake real estate listings in future studies.

## CONCLUSIONS

K-means clustering analysis successfully classified 12,816 unsupervised demographic real estate data lists into two distinct clusters: not fake (7,276) and fake (5,540). Several techniques are used to detect potential patterns and clusters that can distinguish fake or not fake real estate listings. This includes determining the appropriate number of clusters using various techniques such as Elbow, Silhouette Score, Calinski Harbaz Index and Davies Bouldien. As a result, the division into two clusters is the best. This is consistent with the goal of the study to distinguish between fake lists and *vice versa*.

Different techniques such as Camberra, jaccard similarity and overlapping similarity are also used to measure the distance between the clusters. The Camberra technique was found to be suitable for a real estate listing environment where feature selection and linear data are involved. The PCA visualization also shows that the Camberra technique provides a clearer separation of clusters with minimal overlap. The distribution of the data for each group is also more balanced as compared to the expert labels obtained previously, and the balance of the data can avoid errors in the tendency to predict.

Cluster analysis is used to determine the cluster labels. The clusters have significant differences based on attributes such as price, built up sf and expert labels. While the similarity of the two groups is for the attribute of longitude and latitude. This means that the location distribution of the two property groups is in the same area. Then, the performance of the cluster labels is tested with the classification model to evaluate their accuracy, validity and reliability. The accuracy, precision, recall and F1 score values are very good (refer to Table 9). This means that clustering analyzes such as K-means can be used effectively and reliably to forming labels for unsupervised data. However, the use of techniques to determine the number of clusters and distances must be precisely matched to the conditions of the data being analyzed. Then, the labeled data set can be analyzed with a more efficient learning model to produce an optimal classification and prediction for the next study.

In conclusion, clustering analysis emerges as a powerful tool in the fight against fake real estate listings. Using this technique, real estate platforms and authorities can effectively classify and identify fraudulent listings, protecting buyers and sellers from scams. Clustering analysis allows for identifying patterns and similarities among listings, enabling the detection of suspicious properties based on attributes. This process allows outliers that exhibit characteristics deviating from genuine listings to be flagged for further investigation. Moreover, clustering analysis enhances the efficiency of fraud detection by reducing manual efforts involved in identifying fake listings. Real estate platforms can promptly detect and remove fraudulent content by automating this process before it reaches unsuspecting users. Harnessing the potential of clustering analysis empowers the real estate industry to build stakeholder trust. In fact, this cluster data can also be used for further studies combining other multimodal data such as images and text to obtain a more accurate and comprehensive classification from various other forms of data.

### Funding

This research was funded by the Universiti Kebangsaan Malaysia (Grant code: GUP2022-060). The funders had no role in study design, data collection and analysis, decision to publish, or preparation of the manuscript.

### Grant Disclosures

The following grant information was disclosed by the authors:
Universiti Kebangsaan Malaysia: GUP2022-060.

## Competing Interests

Faizal Abd Kadir is the Chief Executive Officer at My Crib Resources.

## Author Contributions

- Maifuza Mohd Amin conceived and designed the experiments, performed the experiments, analyzed the data, performed the computation work, prepared figures and/or tables, authored or reviewed drafts of the article, and approved the final draft.
- Nor Samsiah Sani conceived and designed the experiments, authored or reviewed drafts of the article, and approved the final draft.
- Mohammad Faidzul Nasrudin conceived and designed the experiments, authored or reviewed drafts of the article, and approved the final draft.
- Salwani Abdullah conceived and designed the experiments, authored or reviewed drafts of the article, and approved the final draft.
- Amit Chhabra conceived and designed the experiments, authored or reviewed drafts of the article, and approved the final draft.
- Faizal Abd Kadir conceived and designed the experiments, authored or reviewed drafts of the article, and approved the final draft.

## Data Availability

The data related to properties are available in the Supplemental File and was obtained from the Durian property platform (October 2021): https://www2.durianproperty.com.my.

## Supplemental Information

Supplemental information for this article can be found online at http://dx.doi.org/10.7717/peerj-cs.2019#supplemental-information.

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
