# Peer review of "Clustering analysis for classifying fake real estate listings"

_PeerJ Computer Science, doi:10.7717/peerj-cs.2019_

## Round 0.1 · original submission · Major Revisions

I have received the review reports for your paper submitted to PeerJ Computer Science from the reviewers. According to the reports, I will recommend major revision to your paper. Please refer to the reviewers’ opinions to improve your paper. Please also write a revision note such that the reviewers can easily check whether their comments are fully addressed. We look forward to receiving your revised manuscript soon.

**Language Note:** The review process has identified that the English language must be improved. PeerJ can provide language editing services - please contact us at [email protected] for pricing (be sure to provide your manuscript number and title). Alternatively, you should make your own arrangements to improve the language quality and provide details in your response letter. – PeerJ Staff

Reviewer 1 ·

Basic reporting

1. The readability of this paper needs to be improved.
2. The quality of some figures is not well. Please check and improve the quality of the figures.
3. The motivation of this work is not very clear.
4. In the related work section, what is the analysis between these previous works and the proposed method?

Experimental design

1. Why these two numerical attributes do not transfer to label values? Directly using the numerical values may have an over-fitting problem.
2. How to measure the similarity between these label values in these non-numerical attributes?
3. This proposed method finally chooses what similarity measure methods and how to determine the optimal K value for the K-means method.

Validity of the findings

1. In section 4.7, it describes the utilization of three distinct classification methods. However, the results of experiments in Table 8, only discuss two methods, Random forest and decision tree.
2. How to avoid the over-fitting problem in this work?

Additional comments

1. In this work, it focuses on an imbalanced dataset. Does it pre-process this dataset to the balanced dataset?
2. How to handle the missing values in this dataset?
3. In the proposed framework, what methods are used for the feature selection?

Cite this review as

Reviewer 2 ·

Basic reporting

The proposed article is well-written, and the basic reporting fulfills the required standard.

Experimental design

The experimental setup is comprehensively described.

Validity of the findings

The results are validated through analysis.

Additional comments

Using the Malaysian property dataset, Durian Property Listing, a K-Means clustering analysis was performed to categorize unsupervised fake property listings. K-Means Clustering was found to successfully label unsupervised property listing data with excellent accuracy, reliability, and validity through the analysis of this study. K-Means was probably selected due to its interpretability, flexibility, speed, scalability, and capacity to handle unsupervised data and clustering based on distance. These features make it a strong option for categorizing real estate listings, including identifying fraudulent ones, and they can produce results with exceptional validity, accuracy, and reliability.
The proposed study has significant contributions.
However, there needs to be minor improvement as follows:
1) All abbreviations should be defined the first time they appear in the text.
2) The paper organization paragraph should be added at the end of Introduction
3) The English should be improved.

Cite this review as

---

## Round 0.2 · Minor Revisions

Dear Professor:

I have received the review reports for your paper submitted to PeerJ Computer Science from the reviewers. According to the reports, I will recommend minor revision to your paper. Please refer to the reviewers’ opinions to improve your paper. Please also write a revision note such that the reviewers can easily check whether their comments are fully addressed. We look forward to receiving your revised manuscript soon.

Reviewer 1 ·

Basic reporting

1. In related work section, please confirm ref.[18] is which work. Patoli et al.[18] or Phillip et al.[18]? The description of these two works is different.
2. Why choose the K-Means clustering algorithm to do this work? It needs to show the motivation about what reasons to solve this problem.
3. In Table 7 and Table 8, these two tables are all the same. Please check and confirm it.

Experimental design

1. In section 3.1, how to determine the missing values of which attributes are replaced NA or -1? Which attributes with missing values are removed?

Validity of the findings

1. In the conclusion section, it shows that the accuracy, precision, recognition and F1 score are good in line 483. However, it does not show any experiment results in the results section. How to get this conclusion for this work?

Cite this review as

Reviewer 2 ·

Basic reporting

No comment

Experimental design

No comment

Validity of the findings

No comment

Additional comments

The authors have thoroughly addressed my concerns.
Therefore, the paper is accepted at this stage.

Cite this review as

---

## Round 0.3 · accepted · Accept

The authors have fully addressed the comments of the reviewers. I am happy to make a decision of acceptance to the paper.

Reviewer 1 ·

Basic reporting

1. In the related works section, the last paragraph above the materials & methods section, it shows that several previous works [20-26] used the K-means method to solve complex fraud tactics. However, it only discusses ref. [21], how about the main concept or solving problems of other works.

Experimental design

no comment

Validity of the findings

no comment

Cite this review as